# Microbiological Quality and Safety of Traditional Raw Milk Cheeses Manufactured on a Small Scale by Polish Dairy Farms

**DOI:** 10.3390/foods11233910

**Published:** 2022-12-04

**Authors:** Constantine-Richard Stefanou, Beata Bartodziejska, Magdalena Gajewska, Anna Szosland-Fałtyn

**Affiliations:** Instytut Biotechnologii Przemysłu Rolno-Spożywczego—Państwowy Instytut Badawczy/Institute of Agriculture and Food Biotechnology—State Research Institute, Department of Food Quality, ul. Rakowiecka 36, 02-532 Warszawa, Poland

**Keywords:** raw milk cheese, artisanal cheese, food safety, composition, *Listeria monocytogenes*, *Salmonella* spp., Poland

## Abstract

Polish raw milk artisanal cheese may pose a threat to consumer safety due to pathogen presence. The aim of this study was to assess the microbiological safety, quality and physicochemical composition of cow’s and goat’s milk fresh cheeses produced by farmers on a small scale. A total of 62 samples of six cheese types were analyzed for *Listeria monocytogenes*, *Salmonella* spp., lactic acid bacteria and coliform presence and concentration levels. The physicochemical analysis estimated energy, water, protein, fat, carbohydrate, ash and salt content. The cheeses were also tested for heavy metal contamination. *Listeria monocytogenes* and *Salmonella* spp. were not detected in any of the samples. Coliforms were present in all the goat’s milk cheeses and only in two of the cow’s milk cheeses. Low levels of cadmium, below 0.008 ppm, were detected in three of the cows’ milk samples. The raw milk cheeses studied were free of the pathogens examined and were of high nutritional value.

## 1. Introduction

Among European countries, Poland remains an important producer and a major consumer of milk and dairy product, especially quark cheeses. It is forecast that, in 2023, Polish cheese consumption will reach 18.46 kg per capita [1]. The Polish dairy industry is dominated by white fresh cheeses. The most commonly consumed quark cheese is twaróg, which is available commercially and also made by households.

Twaróg is a Polish fresh acid-set cheese traditionally produced from unpasteurized cow’s milk. The cheese is produced by the natural acidification of milk through endogenous lactic acid microflora for approximately 3 days. The acidified milk is then heated to 43 °C in order for the curds to set and separate. The mixture is then strained for 10 min to remove excess whey and transferred to cold storage for 3 h. After cooling, the fresh cheese is removed from the cheese cloth and packaged. This type of twaróg from unpasteurized milk is commonly produced by small-scale farms and sold on a local level by producers directly to customers. Commercial versions produced on an industrial scale use pasteurized milk and standardized bacterial starter cultures to guarantee consistency of the end product [2].

“Ser podpuszczkowy dojrzewający”, is a traditional fresh cheese produced with the addition of rennet. Approximately 40 mL of natural rennet and 100 mL of whey (per 20 L of milk) are added directly to unpasteurized milk after milking at a temperature of approximately 30 °C. The rennet causes the curd to set and separate from the whey, due to the destabilization of the casein micelles. The set curd is cut into smaller pieces after 30 min to allow for further whey expulsion. After 30 min, the curds are strained from the whey, washed with hot water and massaged before being placed in a sieve to firm up and expel additional water for 10 min. Following this, the cheese is placed in a mold to take shape for 6 h. After production, the cheese is submerged in brine and left to mature for up to four days. Several types of nuts, herbs and spices are traditionally used in the production of this type of cheese, such as fenugreek, wild garlic, chili, paprika, walnuts, dried tomato, nigella seeds, dried onion, horseradish and pepper. These are mixed into the cheese mass and sprinkled on the exterior to provide additional flavor. “Wędzony ser podpuszczkowy dojrzewający” is a variety of rennet cheese, made from unpasteurized milk. Its unique taste is a result of its simple production method that does not require complicated tools and the smoking, for which alder wood is used.

Consumer interest in small-scale locally produced cheeses, such as the aforementioned, has been increasing, with many Poles seeking an alternative to mass-produced industrial dairy products [3]. Because of the growing popularity, small-scale cheese making farms are flourishing. The microbiological quality of such products has not been critically assessed in Poland in relation to the requirements set out in Regulation (EC) No. 852/2004, Regulation (EC) No. 853/2004 and Regulation (EC) No. 2073/2005, which specify the provisions for the production hygiene of animal-origin foods and the microbiological safety criteria required to achieve the set Appropriate Level of Protection (ALOP) for the final consumer. The use of natural raw materials and no additives in the production of such cheeses, alongside the relatively small scale of production, do not necessarily guarantee the appropriate microbiological quality expected by consumers. Furthermore, such cheeses produced from raw milk raise concerns over safety [4,5]. Foods produced locally, often using traditional methods, are rarely controlled in terms of safety, and the farmers/producers themselves often lack knowledge related to good hygiene and production practices. Products of animal origin can be a very good substrate for the growth of microorganisms. Specifically for fresh cheeses, the high nutrient content, high moisture, low-temperature heat treatment and lack of additives conduce to the growth of various microorganisms, including possibly dangerous pathogens that could pose a threat to consumers of these products in Poland. The aim of this study was to assess the general microbiological quality of such products and ascertain their safety in regard to *Listeria monocytogenes* and *Salmonella* spp.

According to EFSA’s 2020 Zoonoses report the prevalence of *Listeria monocytogenes* in cheeses produced from raw or low-heat-treated (LHT) milk ranged between 0.67% and 1.4% of tested samples of soft and semi-soft cheeses. The recorded prevalence of *Listeria monocytogenes* was comparable in dairy products produced from unpasteurized milk and heat-treated milk, for the 2017–2020 period [6]. The Rapid Alert System for Food and Feed (RASFF) is a European system for reporting food safety issues within the Union and is a key feature of EU food safety. In 2015, there were 24 reported cases of *Listeria monocytogenes* in connection with cheeses mostly from France (18, most often reported to be made from raw milk) followed by Italy (6, gorgonzola) [7]. In 2016, nine notifications for *Listeria monocytogenes* were reported in France as a result of companies’ self-testing regimes, resulting in withdrawals. The products in question were often raw milk cheeses [8]. Similarly, in 2017, there were 11 notifications for *Listeria* incidence in French cheese, whilst in 2018, these increased to 13 [9,10]. In 2019, 25 notifications were reported for *Listeria monocytogenes* in milk and milk products [11]. The RASFF report for 2020 included 3 alerts regarding raw milk cheeses out of a total of 25 *Listeria*-cheese notifications [12]. The Polish national food safety authority (Główny Inspektorat Sanitarny) has enforced six recalls of cheeses due to detection of *Listeria* from 2019 to 2021 [13]. These included cheeses made from raw milk and traditional products. *Listeria* is not the most common foodborne pathogen reported, but the small dose required for illness and the severity of listeriosis in humans warrants high vigilance for all RTE foods including cheeses [6]. In the case of traditional raw milk, farm-produced cheeses the risk of listeriosis is increased due to the lack of pasteurization and the level of hygiene maintained on site. Pyz-Łukasik et al. studied the occurrence of *Listeria monocytogenes* in artisanal Polish cheeses and found the prevalence to be much higher than the European average reported by EFSA. Their findings suggest a prevalence of 6.2% for artisanal cheeses, thus highlighting the importance of monitoring for *Listeria* presence in such products [14].

Salmonellosis is the second most commonly reported zoonosis in the EU/EE and has been in some cases linked to cheese consumption [6]. Monitoring of *Salmonella* spp. in the food chain is well established on an EU level. *Salmonella* was found in 0.14% of cheeses, compared with 0.21% in 2019 [6]. In 2015, an alert was issued regarding *Salmonella* Enteritidis presence in a French raw milk cheese, and an outbreak of salmonellosis in 2018 was linked to *Salmonella* Newport detected in French goat cheese made of raw milk [7,10]. For the period of 2020–2021, seven notifications were posted through the RASFF portal for *Salmonella* spp. in cheese, three of these involved raw milk cheeses. It is worth noting that EU Regulation 2073/2005 sets out a microbiological safety criterion for *Salmonella* spp. only in raw milk cheeses.

An important parameter affecting the overall quality of artisanal/traditional dairy products is their indigenous lactic acid microflora, which is characteristic of these cheeses. The lactic acid bacteria (LAB) associated with small-scale local dairy production is often unique and provides the organoleptic properties that distinguish such cheeses from their industrially mass-produced counterparts. Furthermore, some LAB strains have been identified as possible probiotic microorganisms [15]. Certain LAB can act as an additional hurdle against pathogens such as *Listeria* by acidifying the food matrix via lactic acid production, competing for resources and producing antimicrobial peptides such as bacteriocins. The effect of certain LAB on *Listeria monocytogenes* in cheese has been studied by researchers with the prospect of use as an antimicrobial factor [16,17]. Furthermore, there is a trend for isolating specific indigenous LAB strains from traditional products and cheese dairies for commercial use. The produced cheeses are richer in volatile compounds as a result of the native microflora. Such cultures are desired in cheesemaking particularly for their ability to produce flavorful, unique products favored by consumers [4,18].

## 2. Materials and Methods

### 2.1. Sample Collection

A total of 62 cheese samples were collected from farm shops in the Łódź Voivodeship in Poland. Cheeses were purchased directly from farmers, local farm shops and farmer’s markets. The cheeses sampled were all made from raw milk and included both cow’s milk cheeses and goat’s milk cheeses. The product types chosen were acid-set cheese “sery twarogowe” (9 from cow’s milk and 8 from goat’s milk), ripened rennet cheese “sery podpuszczkowe dojrzewające” (11 from cow’s milk and 12 from goat’s milk) and smoked rennet cheese “wędzone sery podpuszczkowe dojrzewające” (12 from cow’s milk and 10 from goat’s milk). The cheeses were stored under refrigeration (T = 0–4 °C) and analyzed at the end of their assigned shelf life, declared by farmers.

### 2.2. Microbiological Analysis

All analytical procedures were performed with 3 replicates for each sample, resulting in a total of 186 analytical samples. For the microbiological analyses, the initial suspension was made with 25 g of sample mechanically homogenized (Stomacher) with 225 mL of buffered peptone water (BPW). Then, decimal dilutions were prepared with a sterile diluent (BPW) and used for plating 1000 μL onto selective mediums. Mesophilic LAB were counted after incubating inoculated de Man, Rogosa and Sharpe agar (MRS) plates at 30 °C for 72 h according to PN-ISO 15214:2002. Coliform bacteria were cultured on violet red bile lactose agar (VRBL), incubated at 37 °C for 24 h according to ISO 4832:2006. The analysis for the presence of *Salmonella* spp. was performed in accordance with PN-EN ISO 6579-1. This includes pre-enrichment in BPW and selective enrichment in Rappaport Vassiliadis Salmonella broth (RVS) and Muller-Kauffmann tetrathionate-novobiocin broth (MKTTn). Streaking from both broths was performed on xylose lysine deoxycholate agar (XLD) and Hektoen agar plates and incubated at 37 °C for 24 h. The detection of *Listeria monocytogenes* was performed in accordance with PN-EN ISO 11290-1. For the analysis, 25 g of cheese was stomached in 225 mL of pre-enrichment broth (half Fraser broth) and incubated at 30 °C for 24 h. This was followed by a secondary selective enrichment broth (Fraser broth) incubated at 37 °C for 24 h. The enriched samples were cultured on agar listeria according to Ottaviani and Agosti (ALOA) and Palcam agar plates at 37 °C for 48 h and checked for signs of growth after 24 h. All media were purchased from Argenta, Oxoid.

### 2.3. Physicochemical Analysis

Physicochemical analytical methods were used to ascertain the chemical and nutritional profile of the farm cheeses. Water and fat content were determined according to the PN-73/A-8623 standard of “PKN”, the Polish Committee for Standardization and Measures. Protein content was determined with the Kjeldahl method (Standard PN-75/A-04018/Az3:2002) and total ash based on PN-ISO 936:2000 gravimetric method. Carbohydrates were calculated as the difference between the sample weight and the estimated water, fat, protein and ash as per Regulation (EU) No. 1169/2011. The macronutrients were used for calculating the calorific value in kcal and kJ per 100 g. The composition of free fatty acids (FFAs) in the cheeses was analyzed using gas chromatography with flame ionization detection (GC-FID) according to an internal standard of the IBPRS-PIB accredited laboratory (ds/PA/05 test procedure 6 of 18 April 2017) [19]. All analyses were performed with 3 replicates for each sample, and final results were expressed in mass percent concentration (% m/m). For the chemical risk assessment of the products, heavy metal analysis was performed for mercury (Hg), cadmium (Cd) and lead (Pb), and results were compared with the legal contamination limits. The AMA 254 mercury analyzer (LECO Korea co. Ltd., Seoul, Korea) was used based on the mercury vapor generation technique according to test procedure PS-02 edition 3 of 6 July 2009, developed by the Food Quality Department of IBPRS-PIB [19]. The detection limit for Hg according to this method was 0.02 mg/kg. Cadmium and lead were estimated with flame atomic absorption spectrometry (FAAS) according to ISO standard PN-EN 14082:2004, with a detection limit of 0.003 and 0.001 mg/kg, respectively. Similarly, the salt content was estimated via FAAS determination of sodium and multiplying the result with a conversion factor of 2.5.

### 2.4. Statistical Analysis

Statistical analysis of the results was performed using Microsoft Office 2016 Excel (Redmond, WA, USA).

## 3. Results

### 3.1. Microbiological Quality

Table 1 and Table 2 display a summary of the microbiological results for the cow’s and goat’s milk cheeses, respectively, while the raw data are presented in the Appendix A (Table A1). All samples exhibited pathogen absence for *Listeria monocytogenes* and *Salmonella* spp. For the bovine milk cheeses, coliforms were only detected in the acid-set cheese. Prevalence was 22%, with two out of the nine cow milk samples having a mean concentration of 5.35 $\log_{10}$CFU/g. In the goat milk cheeses, all types had coliform presence, with mean concentration values ranging from 2.34 to 8.23 $\log_{10}$CFU/g. Coliforms were detected in 5/8 of the acid-set cheese samples, 6/12 of the matured rennet cheese and 3/10 of the smoked rennet cheese samples.

Lactic acid bacteria concentrations were more similar in comparison among samples. Between the two product categories, goat milk cheeses had slightly higher LAB mean values than cow milk cheeses, which displayed a bigger range in concentrations. The *t*-test performed of the LAB results found the smoked rennet cheese to have a statistically significant difference in LAB concentrations between cow milk and goat milk cheese.

### 3.2. Physicochemical Properties

To assess the nutritional aspect of these cheeses, the energy, moisture, macronutrient, salt and ash content results are presented in Table 3 and Table 4. The mean values for protein, carbohydrate, ash and salt were close between cow’s milk and goat’s milk cheeses. Mean calorific content for all the farm cheeses ranged from 145.00 kcal/100 g up to 387.00 kcal/100 g. Goat milk cheeses had lower energy values in comparison with their cow milk counterparts, reflecting their lower fat content. The biggest difference was in the smoked rennet cheese, where the mean energy values were 366.00 kcal/100 g for cow’s milk cheese and 266.00 kcal/100 g for goat’s milk cheese. Moisture levels were similar for each individual product type regardless of milk origin. The statistical significance of the noted differences between milk types for each parameter were examined using the independent samples *t*-test (*p* < 0.05). For the acid-set cheese, differences in protein, fat and salt content between bovine and caprine origins were found to be statistically significant. The matured rennet cheese displayed a statistically significant mean difference for fat, moisture, ash and energy. For the smoked rennet cheese, the mean differences where statistically significant for all parameters except carbohydrates.

In 3/9 of the cow’s milk acid-set cheese samples (33.33%), low levels of cadmium, below 0.008 mg/kg, were detected. The rest of the samples were clear of heavy metal contamination of cadmium, lead and mercury.

## 4. Discussion

The lack of pathogens in these types of products is a multicomponent result, stemming from product composition, innate microbial competition and process environment hygiene. Fresh cheeses have low pH values, which acts as a hurdle to pathogen growth. Furthermore, the lack of heat treatment means certain natural milk components are able to act as inhibitors. The presence of endogenous enzymes, which would have been denatured if the milk was pasteurized, could be a factor providing a protective effect. The antagonistic ability of LAB against pathogens is a very important hurdle for cheese safety and is the result of cell mechanisms and competition in the food matrix. Among these are the acidification of the product, lactic acid production, antimicrobial compounds and competition for nutrients. As with all food production, the maintenance of high hygiene standards and codes of production practice contribute toward food safety goals. *Listeria* was not detected in any of the samples. In their study, Maćkiw et al., who sampled 22,842 milk and dairy products in Poland, found no presence of *Listeria monocytogenes* in cheese [20]. On the contrary, Pyz-Łukasik et al.’s study on Polish artisanal rennet cheeses estimated a mean prevalence of 6.2% for *Listeria monocytogenes*. *Salmonella* spp. was not detected in any of the tested cheeses, so any presence would be below the methods’ detection limit. Łobacz et al. studied the survival of *Salmonella* in raw milk twaróg cheese during storage and concluded that growth in this product is inhibited and that storage temperature is the main parameter influencing the inactivation rate. *Salmonella* spp. survived in lower temperatures and inactivation rates increased as storage temperature increased [4]. Coliforms have been used by the dairy industry as a hygiene indicator organism for milk products. The diversity of this bacterial group has led to questioning of their suitability as an indicator. In the case of raw milk cheeses, their prevalence in unpasteurized milk means that their detection in the final product is not surprising [4,21]. Jackson et al., who studied the relationship of coliform bacteria concentrations and pathogens (*Bacillus cereus*, *Escherichia coli* O157:H7, *Listeria monocytogenes* and *Salmonella* spp.) in raw milk, concluded that there was no significant correlation between coliform levels and pathogen presence [22]. This variability is why the European Regulation 2073/2005 does not consider coliforms in regard to milk products and sets coagulase-positive staphylococci as a risk-based hygiene criterion for raw milk cheeses. Possible future research with these types of products could include the specific EU hygiene criteria.

The production of cheeses from unpasteurized milk is not so popular owing to the potential problems with safety. Therefore, the studies about microbial and physicochemical quality in the literature for raw milk cheeses are limited [23]. In this research, physicochemical characterization of Polish, artisanal cheeses made from cow’s and goat’s milk was presented. The largest amounts of fat were observed in matured rennet cheeses manufactured from cow’s milk, whereas the smallest was in goat’s milk “twarogi”. Matured rennet cheeses also had higher values for protein content, in comparison to acid cheeses. The obtained results are in consistence with Ioannidou et al. The authors stated that, during the ripening time, protein and fat contents increase because of the dehydration process [23].

Environmental pollution is one of the most significant factors contributing to the occurrence of many toxic metals in foods. Cadmium, mercury and lead in particular are considered highly toxic and have harmful effects on human health. For this reason, the determination of the heavy metal level in cheeses is very important [24]. As for the heavy metal concentration in this study, their content was very low, comparable to values reported in the literature [24,25,26,27]. Lead was below the permissible levels. It is worth highlighting that this element tends to associate with casein, which may contribute to an increase of its concentration in cheeses [24]. According to European Regulation 1881/2006 about food contaminants, there are no specific limits for cadmium and mercury in milk and dairy products. In the present study, low levels of cadmium, below 0.008 ppm, were detected. This is, nevertheless, important due to the high toxicity presented by cadmium [26]. Our study demonstrated a concentration of cadmium similar to the level obtained by Al Sidawi et al. The most often indicated source of cheese contamination with toxic metals by these authors is residues of these metals in animal fodder and in water [24].

## 5. Conclusions

The market for artisanal, traditional cheeses from raw milk continues to expand worldwide. The results from this study demonstrated that all studied samples met the existing legal microbiological and physicochemical parameters. The tested cheeses were deemed of high nutritional value and quality. Although none of the samples tested contained *Salmonella* spp. or *Listeria monocytogenes*, isolation of coliforms from different kind of cheeses suggests the need to focus attention on hygienic procedures during the production stages. In practice, increased awareness of the importance of hygienic practice by Polish farmers is required. Regular monitoring of these kind of products is necessary to ensure the health safety of consumers.

## Figures and Tables

**Table 1 foods-11-03910-t001:** Cow’s milk cheese microbiological results.

	Acid-Set Cheese				Matured Rennet Cheese				Smoked Rennet Cheese			
Parameter	Min.	Mean	SD ^2^	Max.	Min.	Mean	SD ^2^	Max.	Min.	Mean	SD ^2^	Max.
LAB ^1^ ($\log_{10}$CFU/g)	6.79	7.88	0.60	8.71	6.40	7.85	0.77	8.72	6.54	6.91	0.19	7.28
Coliforms ($\log_{10}$CFU/g)	5.00	5.35	0.29	5.65	<10				<10			
*Listeria monocytogenes*	Not detected				Not detected				Not detected			
*Salmonella* spp.	Not detected				Not detected				Not detected			

^1^ Lactic acid bacteria. ^2^ Standard deviation of the mean.

**Table 2 foods-11-03910-t002:** Goat’s milk cheese microbiological results.

	Acid-Set Cheese				Matured Rennet Cheese				Smoked Rennet Cheese			
Parameter	Min.	Mean	SD ^2^	Max.	Min.	Mean	SD ^2^	Max.	Min.	Mean	SD ^2^	Max.
LAB ^1^ ($\log_{10}$CFU/g)	7.11	7.92	0.56	8.65	7.46	8.14	0.38	8.66	7.78	8.11	0.22	8.23
Coliforms ($\log_{10}$CFU/g)	2.34	3.70	1.53	5.68	5.00	5.73	0.4	6.23	6.32	6.97	1.00	8.23
*Listeria monocytogenes*	Not detected				Not detected				Not detected			
*Salmonella* spp.	Not detected				Not detected				Not detected			

^1^ Lactic acid bacteria. ^2^ Standard deviation of the mean.

**Table 3 foods-11-03910-t003:** Cow’s milk cheeses’ physicochemical results.

		Acid-Set Cheese				Matured Rennet Cheese				Smoked Rennet Cheese			
Parameter		Min.	Mean	SD	Max.	Min.	Mean	SD	Max.	Min.	Mean	SD	Max.
Energy	kcal/100 g	138.00	160.00	13.00	185.00	281.00	316.00	16.00	361.00	303.00	366.00	14.00	387.00
	kJ/100 g	572.00	666.00	28.00	769.00	1166.00	1314.00	53.00	1494.00	1264.00	1425.00	44.00	1620.00
Water% m/m		68.20	71.90	2.20	75.00	45.90	50.60	2.40	56.60	42.80	43.80	1.30	45.10
Protein% m/m		11.20	12.50	1.00	13.70	17.80	19.60	1.20	21.50	20.00	21.30	0.70	22.80
Total Fat% m/m		10.00	12.10	0.80	14.20	22.80	25.80	1.90	30.00	25.40	26.70	1.10	30.30
Saturated Fats% m/m		6.80	8.20	0.70	9.60	15.30	17.40	1.00	20.30	17.00	17.90	1.00	20.30
Carbohydrates% m/m		0.50	0.90	0.20	1.20	1.00	1.60	0.50	2.10	1.00	1.20	0.40	1.50
Ash% m/m		2.80	3.20	0.20	3.30	2.80	3.10	0.30	3.30	3.10	3.20	0.80	3.30
Salt% m/m		0.05	0.07	0.01	0.09	1.10	1.38	0.13	2.12	1.36	1.55	0.12	2.02

**Table 4 foods-11-03910-t004:** Goat’s milk cheeses’ physicochemical results.

		Acid-Set Cheese				Matured Rennet Cheese				Smoked Rennet Cheese			
Parameter		Min.	Mean	SD	Max.	Min.	Mean	SD	Max.	Min.	Mean	SD	Max.
Energy	kcal/100 g	145.00	149.00	11.00	155.00	257.00	272.00	15.00	303.00	254.00	266.00	11.00	279.00
	kJ/100 g	604.00	620.00	20.00	647.00	1069.00	1128.00	56.00	1256.00	1056.00	1106.00	48.00	1156.00
Water% m/m		70.90	71.30	2.00	73.60	50.10	55.20	2.00	56.30	54.60	55.00	1.50	56.50
Protein% m/m		12.50	13.30	1.20	15.20	17.70	19.90	1.50	21.60	17.90	19.00	1.30	20.50
Total Fat% m/m		10.30	10.80	0.90	11.20	20.00	21.50	1.80	24.30	18.10	20.30	1.10	22.50
Saturated Fats% m/m		6.80	7.00	0.70	7.40	14.00	15.10	0.90	17.50	14.10	15.80	1.50	17.60
Carbohydrates% m/m		0.30	0.80	0.20	1.10	0.20	1.00	0.40	2.10	0.80	1.80	0.50	2.80
Ash% m/m		3.00	3.20	0.20	3.40	3.00	3.40	0.50	4.10	3.10	3.30	1.50	3.60
Salt% m/m		0.05	0.05	0.01	0.06	1.17	1.32	0.14	1.48	1.28	1.41	0.15	1.51

## Data Availability

Data is contained within the article.

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
