# Peer review of "Microbiological Quality and Safety of Traditional Raw Milk Cheeses Manufactured on a Small Scale by Polish Dairy Farms"

_foods, 2022, doi:10.3390/foods11233910_

Round 1

Reviewer 1 Report

The manuscript “Microbiological quality and safety of traditional raw milk cheeses manufactured on small scale by Polish dairy farms” aims to characterize 6 different types of artisanal cheese from Poland in relation to the microbiological quality. Some specific comments:

-       the introduction presents relevant information, but it is too extensive, mainly when compared with the section “Discussion”, which is rather reduced and could be further explored;

-       The text has to be revised. There are several typographic and language errors (e.g. lines 57, 62, 102, 119, 223, 249, …)

-       Is there any reason to focus on data from the European Union One Health 2019 Zoonoses Report (several parts of introduction and Tables 1 and 2) when there is already a more recent version of 2020?;

-       Table 2 (page 3) is not very useful – it contains just one line and therefore the most relevant data can be put on text;

-       Some names of microorganisms are not in italic (example: line 155, Listeria monocytogenes…);

-       Line 143 – as described by the authors, cheeses were obtained from local markets, farmers etc. Do the cheeses from farmers had the indication of shelf life? This might be relevant because the comparison between cheeses might not make any sense because they were analysed under potentially different premises about the shelf life… 

-       The Materials and Methods section should be revised  - some culture media and diluents are described just in an abbreviated form and the commercial brand used is not presented (ex: BPW, MKTT, MRS…);

-       Lines 147 and 157 – “…stomached…” – mechanically homogenized (Stomacher)?

-       Line 182 – units?

-       There are two Tables 2 (page 3 and page 5). In the Table 2 from page 5, the results for the parameter “coliforms” is presented as Not Detected for the Matured Rennet Cheese and for the Smoked Rennet Cheese. This is not correct. Coliforms were quantified. As it was not a detection, but a quantification, the result should be presented as < to the detection limit.

-       Lines 204-205 – p value of the t-test?

-       Line 223 – results of the statistical analysis?

Author Response

Dear Reviewer,

We appreciate you for your precious time in reviewing our paper and providing valuable comments. We have carefylly considered the modifications, that led to possible improvements in the current version.  We revised the Introduction, the Materials and Methods and Discussion section. We hope the manuscript after our revisons meet your higher standards. We welcome further constructive comments if any.

Sincerely,

Anna Szosland- Fałtyn, PhD

Reviewer 2 Report

The manuscript titled” Microbiological quality and safety of traditional raw milk cheeses manufactured on small scale by Polish dairy farms” was reviewed for consideration in foods. The topic is interesting and worth useful for researchers and scientists working in this field. The work carried out in systematic and logical manner. However few observations were notices which should be incorporated during revision.

Introduction

Line 23: Cheese consumption cited in reference 1 is of 2005, should be of latest years.

Line 36 and 50 started with same words, should be changed, it produces repetition during paragraphs.  

Line 135: The sampling location should be presented in map; it creates more visibility and clear picture of the locations of samples.

Table 2 and Table 3: stdv should be changed as “SD”

Line 220: Please quote exact p vale if possible

Conclusion

Please add conclusion part, its very important to discuss the main finding and further recommendation in conclusion part

Author Response

Dear Reviewer,

We appreciate for your precious time and valuable comments. We have carefully considered all modifications. We hope the manuscript after revisions meet your high standards.  We welcome further constructive comments if any.

Sincerely,

Anna Szosland-Fałtyn, PhD
